# Speculative Decoding: Exploiting Speculative Execution for Accelerating Seq2seq Generation

**Heming Xia**[1,2][*][†]  **Tao Ge**[4][†]  **Peiyi Wang**[1,3]  **Si-Qing Chen**[4]  **Furu Wei**[4]  **Zhifang Sui**[1,3]

[1]National Key Laboratory for Multimedia Information Processing, Peking University
[2]School of Software & Microelectronics, Peking University
[3]School of Computer Science, Peking University   [4]Microsoft Research Asia
{xiaheming,szf}@pku.edu.cn; wangpeiyi9979@gmail.com
{tage,fuwei}@microsoft.com

## Abstract

We propose Speculative Decoding (SpecDec), for the first time ever[1], to formally study exploiting the idea of speculative execution to accelerate autoregressive (AR) decoding. Speculative Decoding has two innovations: Spec-Drafter – an independent model specially optimized for efficient and accurate drafting – and Spec-Verification – a reliable method for verifying the drafted tokens efficiently in the decoding paradigm. Experimental results on various seq2seq tasks including machine translation and abstractive summarization show our approach can achieve around $5\times$ speedup for the popular Transformer architectures with comparable generation quality to beam search decoding, refreshing the impression that the *draft-then-verify* paradigm introduces only $1.4\times\sim2\times$ speedup. In addition to the remarkable speedup, we also demonstrate 3 additional advantages of SpecDec, revealing its practical value for accelerating generative models in real-world applications. Our models and codes are available at https://github.com/hemingkx/SpecDec.

## 1 Introduction

As the *de facto* method for text generation, AutoRegressive (AR) decoding is widely blamed for its poor inference efficiency due to its low level of parallelism, which fails to utilize the full potential of modern parallel computing devices like GPUs. This inefficiency not only leads to high deployment

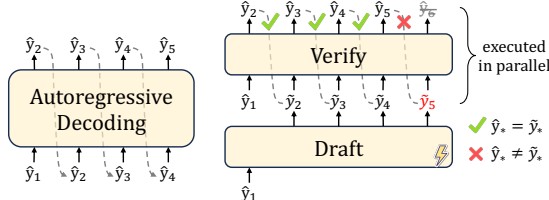

Figure 1: Compared with autoregressive decoding (*left*) that generates token by token, the *draft-then-verify* paradigm (*right*) first *drafts* multiple tokens efficiently and then *verifies* these tokens in parallel. Drafted tokens after the bifurcation position (*e.g.*, $\widetilde{y}_5$) will be discarded to guarantee the generation quality.

costs but also limits the application of advanced AR models in real-time scenarios.

In this work, we study the *draft-then-verify* paradigm for accelerating seq2seq generation of an existing AR model[2]. As shown in Figure 1, the *draft-then-verify* paradigm first generates a number of drafted tokens efficiently and then verifies these tokens using the existing AR model in parallel to ensure the decoding result matches AR decoding. However, previous attempts in the "*draft-then-verify*" paradigm such as Blockwise Decoding (Stern et al., 2018) and Aggressive Decoding (Sun et al., 2021) tend to lack in-depth investigation of this paradigm. Their modest speedup (i.e., $1.4\times\sim2.0\times$) or limitation to certain seq2seq tasks like Grammatical Error Correction (GEC) has caused this paradigm to be underestimated, resulting in it not receiving much attention and remaining dormant for years.

To fully exploit the *draft-then-verify* paradigm, we propose Speculative Decoding (SpecDec), drawing inspiration from speculative execution[3] in com-

---

[*] This work was done during the author's internship at MSR Asia. Correspondence: Tao Ge (tage@microsoft.com)

[†] Co-first authors with equal contributions

[1]This work was initially announced in **March 2022** (https://arxiv.org/abs/2203.16487) under the name *Generalized Aggressive Decoding*. It has been formally renamed *Speculative Decoding* in our submission to ICLR'23, which has been publicly available since **September 2022** at https://openreview.net/pdf?id=H-VlwsYvVi. This marks **the first time** "Speculative Decoding", which explicitly studies the idea of speculative execution to accelerate Transformer inference, has been proposed.

[2]The existing AR model in this paper refers to the targeted Transformer using AR decoding that we want to accelerate.

[3]Speculative execution is an optimization technique used in computer architecture where a system performs some task in advance to avoid delays that would have to be incurred by doing the task after it is known that it is required (https://wikipedia.org/wiki/Speculative_execution).

puter architecture, with two key innovations that improve drafting and verification processes respectively. For drafting, we derive two principles for designing the drafting model[4]: *the Capability Principle* and *the Latency Principle*. Following these two principles, we propose Spec-Drafter – a specialized independent model optimized in the *draft-then-verify* paradigm, which can accurately and efficiently fulfill the drafting task.

For verification, we propose an advanced method – Spec-Verification that relaxes the vanilla verification strategy. Spec-Verification allows the decoding results of SpecDec to be slightly different from AR greedy decoding, offering an opportunity to accept more drafted tokens without sacrificing generation quality and leading to higher decoding efficiency.

We conduct extensive experiments on various seq2seq generation tasks like machine translation and abstractive summarization. Results show our approach can achieve around $5\times$ speedup for the popular Transformer architectures with comparable generation quality to beam search decoding, largely outperforming previous *draft-then-verify* work ($1.4\times\sim2.0\times$ speedup). Moreover, we demonstrate that SpecDec has several additional advantages that enhance its practicality for accelerating generative models in real-world applications.

Our contributions can be summarized as follows:

- We are the first work that explicitly exploits the idea of speculative execution to accelerate Transformer inference. Our proposed two key innovations – the independent Spec-Drafter and Spec-Verification strategy allow SpecDec to achieve over $5\times$ lossless speedup over autoregressive decoding in seq2seq tasks, refreshing the impression that the "*draft-then-verify*" paradigm only has a limited $1.5\times\sim$ $2\times$ acceleration potential.

- We demonstrate 3 advantages of SpecDec with extensive empirical results in addition to its remarkable acceleration performance: better latency-throughput trade-off, easy adaptability for existing models and retaining the behavior of the original model, revealing its huge practical value and bringing the long-dormant *draft-then-verify* paradigm back into the spotlight.

---

[4]The drafting model is also called the drafter in this paper.

## 2 Background: *draft-then-verify* decoding

The "*draft-then-verify*" paradigm first *drafts* multiple tokens efficiently, as a *speculation* of AR decoding results; then, it *verifies* these tokens in parallel to ensure they match the AR decoding result, as illustrated in Figure 1. It is an implicit implementation of speculative execution in Transformer inference.

**Draft** There are different approaches to drafting tokens, including model-based (Stern et al., 2018) and input-(context-) based[5] methods (Sun et al., 2021; Yang et al., 2023). Take Blockwise Decoding – the most representative work attempting the *draft-then-verify* paradigm – as an example (illustrated in Figure 2(a)): it introduces additional $k-1$ feedforward network (FFN) heads on top of an *existing* AR model, enabling the model to predict the next $k$ drafted tokens in parallel during inference.

**Verify** The generated drafted tokens are fed into the original AR model and verified in parallel. Specifically, it finds the bifurcation position $c$, the largest index that ensures all previous $c-1$ drafted tokens and the corresponding AR decoded tokens are identical:

$$c = \arg\max_i \frac{\mathbb{I}(\widetilde{y}_{j+i} \neq \hat{y}_{j+i})}{i}, 1 \leq i \leq k \qquad (1)$$

$$\hat{y}_{j+i} = \arg\max_y \log P(y \mid \hat{\boldsymbol{y}}_{\leq j}, \widetilde{\boldsymbol{y}}_{j+1\cdots j+i-1}, \boldsymbol{x}; \boldsymbol{\theta}_{\text{AR}}) \quad (2)$$

where $\mathbb{I}(\cdot)$ is the indicator function, $\boldsymbol{x}$ is the source sentence, $\hat{\boldsymbol{y}}_{\leq j}$ is the previously generated tokens[6] and $\widetilde{y}_{j+i}$ is the $i$-th drafted token. Drafted tokens after the position $c$ are all discarded. The final decoded tokens in the current iteration are:

$$\hat{\boldsymbol{y}}_{j+1\cdots j+c} = (\widetilde{\boldsymbol{y}}_{j+1\cdots j+c-1}, \hat{y}_{j+c}) \qquad (3)$$

The above draft and verification steps are iterated until the termination condition is met.

## 3 Speculative Decoding

To fully exploit speculative execution for Transformer inference, we propose Speculative Decoding (SpecDec) with two innovations – Spec-Drafter and Spec-Verification that substantially improve drafting (Section 3.1) and verification (Section 3.2) respectively.

---

[5]This kind of method is usually limited to special tasks like GEC and retrieval-augmented generation. In this paper, we mainly focus on the model-based methods.

[6]We use $\widetilde{\boldsymbol{y}}$ to denote drafted tokens, while we use $\hat{\boldsymbol{y}}$ to denote AR decoded/verified generation results.

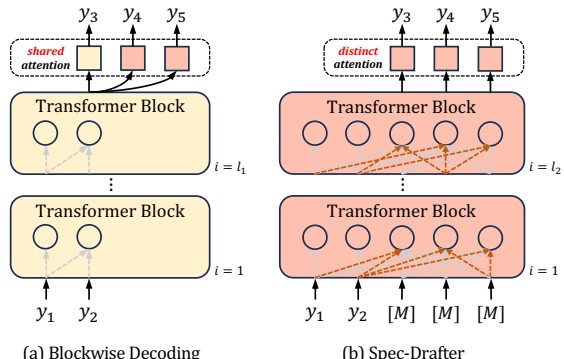

(a) Blockwise Decoding  (b) Spec-Drafter

Figure 2: **(a)** Blockwise Decoding that introduces $k-1$ FFN heads on top of the target AR model for drafting the next $k$ tokens with shared attention; **(b)** Spec-Drafter is an independent model for drafted token prediction. It employs distinct attention queries for predicting each drafted token. Modules colored in yellow belong to the *original* AR model while those colored in red denote newly introduced modules.

## 3.1 Spec-Drafter

### 3.1.1 Design Principles

As a crucial ingredient in the *draft-then-verify* paradigm, the drafting process has a drastic impact on end-to-end acceleration performance. However, there are very limited explorations of the designing principles for the drafter by previous studies – most of them arbitrarily implement a drafter, which accounts for their undesirable acceleration results.

To understand the effect of drafting, we look into the overall latency in the *draft-then-verify* paradigm for one sample of the length $L$ as follows:

$$T = \underbrace{\frac{L}{\textbf{Tok.}} \times t_d}_{\text{total drafting latency}} + \underbrace{\frac{L}{\textbf{Tok.}} \times t_v}_{\text{total verification latency}} \quad (4)$$

where **Tok.** denotes the average number of drafted tokens accepted per iteration, $t_d$ and $t_v$ are the time costs of drafting and verification[7] each iteration respectively.

According to Eq (4), **Tok.** is inversely proportional to the number of iterations, which is primarily influenced by drafting accuracy: A drafter that is more capable of drafting can attain greater **Tok.** values, consequently completing the decoding process in fewer iterations. This observation leads us to derive the first principle for designing the drafter:

**Principle I (Capability Principle)**: The drafter model should be seriously in-

[7]We don't discuss $t_v$ in this paper because it is determined by the existing AR model and thus regarded constant.

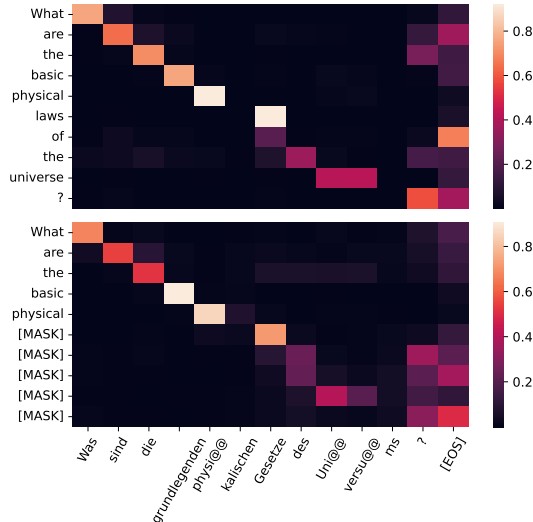

Figure 3: **Upper**: An AR model's attention heatmap showing that different target positions should attend to different source tokens; **Lower**: The Spec-Drafter's attention heatmap showing its capability of modeling drafted tokens in different positions, which highly aligns with the AR counterpart.

vested to guarantee its capability of accurate drafting.

Principle I is the most crucial principle in determining the end-to-end speedup, as it directly influences the value of **Tok.** which affects both total drafting and verification latency. Surprisingly, little previous work adheres to this seemingly simple and straightforward principle maybe due to the concern of increasing the drafting latency. For instance, the drafter in Blockwise Decoding is not properly invested: Its drafter not only has limited parameters (FFN prediction heads), making it difficult to fit the challenging drafting task, but more importantly, it employs a shared attention mechanism that forces all drafted tokens to share a single set of attentions (only differentiating at the final prediction head), as shown in Figure 2(a). However, different target positions should attend different context tokens, as illustrated in Figure 3. Despite its computational efficiency, the shared attention mechanism in Blockwise decoding severely constrains the drafter's capability, resulting in low drafting accuracy and consequently leading to most drafted tokens being discarded.

In addition to the drafter's accuracy, its latency also impacts the end-to-end speedup result but from another perspective – by affecting the latency of each iteration (i.e., $t_d$ in Eq (4)) – from which we derive Principle II for designing the drafter:

| Source Sentence | Machen sich Hunderte Millionen von Autofahrern sorgen über ihre Privatsphäre. |
|---|---|

| Decoder Input | Millions | of | [MASK] | [MASK] | [MASK] | [MASK] | [MASK] | |
|---|---|---|---|---|---|---|---|---|
| Draft | Millions | of | drivers | will | be | concerned | about | |
| Spec-Verify | Millions | of | **drivers** | **will** | worry | **concerned** | **about** | their |
| | | $\beta=3$ | motor@@ | [BLANK] | **be** | [BLANK] | [BLANK] | |
| | | | [BLANK] | [BLANK] | [BLANK] | [BLANK] | [BLANK] | |
| Output | Millions | of | drivers | will | be | concerned | about | their |
| Next Input | Millions of drivers will be concerned about their | | [MASK] | [MASK] | [MASK] | [MASK] | [MASK] | |

Figure 4: Illustration of Spec-Verification. Compared to the vanilla verification strategy strictly requiring the drafted tokens to match the AR top-1 result, Spec-Verification slightly relaxes the criterion to trust the drafts more, by only requiring the drafted tokens to fall in the *top-$\beta$* AR candidates with a tolerable log-likelihood gap (not shown in this Figure; see Eq (9)). As a result, Spec-Verification allows more drafted tokens to be accepted even if they are slightly different from the AR top-1 result, leading to a higher inference speedup.

**Principle II (Latency Principle)**: The drafter should be fast at generating drafted tokens to minimize the latency overhead of each iteration.

Designing a fast drafter solely based on Principle II is not difficult, as done in most previous work. The real challenge lies in designing a low-latency drafter without compromising its capability (Principle I), since it is difficult to achieve both low latency and high capability simultaneously.

### 3.1.2 Model Architecture

We propose Spec-Drafter, which adheres to both principles for accurate and fast drafting. To ensure the drafter is sufficiently capable of accurate drafting (Principle I), Spec-Drafter employs an independent encoder-decoder model architecture, which generates drafted tokens conditioned on the leftward context and source tokens in a mask-predict manner (Ghazvininejad et al., 2019), as illustrated in Figure 2(b). This independent model design facilitates Spec-Drafter to predict each drafted token using distinct attention queries, in contrast to Blockwise Decoding employing a shared attention query for predicting all drafted tokens (as illustrated in Figure 2). In this way, Spec-Drafter could better align with the AR model's behavior, thereby increasing the chances of its drafted tokens being accepted during verification, as shown in Figure 3.

To make Spec-Drafter fast (Principle II) without compromising its capability, we design its decoder to be lightweight by reducing the number of decoder layers and reallocating the freed-up budget to

its encoder (by increasing its depth), which is motivated by the fact that the encoder is forwarded only once, while the decoder is frequently forwarded for iterative decoding. This encoder-favored modeling has been demonstrated by previous work to improve latency with little generation quality degradation (Kasai et al., 2021; Sun et al., 2021; Ge et al., 2022a). We find it also highly effective for the drafter in the *draft-then-verify* decoding paradigm.

### 3.1.3 Training and Inference

Formally, given the source sentence $\boldsymbol{x}$ and the randomly sampled prefix $\boldsymbol{y}_{\leq p}$ ($0 \leq p < m$) of the target sentence, Spec-Drafter appends $k$ special "[MASK]" tokens to $\boldsymbol{y}_{\leq p}$, and is trained to predict these masked tokens in parallel:

$$\mathcal{L}_{\text{Spec-Drafter}} = \sum_{i=p+1}^{p+k} \log P\left(y_i \mid \boldsymbol{y}_{\leq p}^k, \boldsymbol{x}; \boldsymbol{\theta}_{\text{Spec-Drafter}}\right) \quad (5)$$

$$\boldsymbol{y}_{\leq p}^k = (y_1, \cdots, y_p, \underbrace{[\text{MASK}], \cdots, [\text{MASK}]}_{\times k}) \quad (6)$$

In addition, we leverage the glancing strategy following Qian et al. (2021), which exploits curriculum learning during training to get better generation performance.

During inference, Spec-Drafter appends $k$ "[MASK]" tokens to the previously decoded tokens $\hat{\boldsymbol{y}}_{\leq j}$ and simultaneously predict these masked tokens as a drafted block:

$$\widetilde{y}_{j+i} = \arg\max_y \log P\left(y \mid \hat{\boldsymbol{y}}_{\leq j}^k, \boldsymbol{x}; \boldsymbol{\theta}_{\text{Spec-Drafter}}\right) \quad (7)$$

where $i = 1, \ldots, k$.

## 3.2 Spec-Verification

As introduced in Section 2, the vanilla verification strategy of preliminary studies only accepts the drafted tokens that match the top-1 result of the AR model, which guarantees that the decoding results are identical to AR greedy decoding. However, the top-1 results are not necessarily better than the drafted tokens, especially when the paradigm is equipped with a high-quality drafter. Therefore, the strict verification criterion (i.e., top-1 matching) will result in many good drafted tokens being discarded just because they are different from the top-1 result of the AR model, which limits the speedup of the paradigm.

To make better use of the drafting results, we propose an advanced verification strategy named Spec-Verification, which is illustrated in Figure 4. Instead of the rigid matching requirement shown in Eq (2), Spec-Verification relaxes the criterion to trust the drafting results more, by only requiring the drafted tokens to fall in *top-β* candidates with a tolerable (log-likelihood) score gap $\tau$ (away from the top-1 result). Formally, it will accept the $i$-th drafted token $\widetilde{y}_{j+i}$ if all previous $i-1$ tokens are accepted, and Eq (8) and (9) are both true:

$$\log P(\widetilde{y}_{j+i}|\triangle; \boldsymbol{\theta}_{\text{AR}}) \geq \log P(\hat{y}_{j+i}^{(\beta)}|\triangle; \boldsymbol{\theta}_{\text{AR}}), \quad (8)$$

$$\log P(\hat{y}_{j+i}^{(1)}|\triangle; \boldsymbol{\theta}_{\text{AR}}) - \log P(\widetilde{y}_{j+i}|\triangle; \boldsymbol{\theta}_{\text{AR}}) \leq \tau, \quad (9)$$

$$\triangle = \hat{\boldsymbol{y}}_{\leq j}, \widetilde{\boldsymbol{y}}_{j+1 \cdots j+i-1}, \boldsymbol{x}, \quad (10)$$

where $\log P(\hat{y}_{j+i}^{(\beta)}|\triangle; \boldsymbol{\theta}_{\text{AR}})$ is the top-$\beta$ ranked result's log-likelihood score by the AR model.

## 4 Experiments

### 4.1 Experimental Settings

**Datasets and Evaluation** We mainly evaluate our approach on two standard machine translation benchmarks: WMT14 EN↔DE (4.5M pairs) and WMT16 EN↔RO (610K pairs). Following prior work (Ott et al., 2018), for WMT14 EN↔DE translation, we adopt *newstest-13* as our validation set for finding the best hyperparameters, and test on *newstest-14*. For WMT16 EN↔RO translation, we use the dataset released by Lee et al. (2018), where *newsdev2016* and *newstest2016* are taken as validation and test sets. We use 32K Byte Pair Encoding (BPE) (Sennrich et al., 2016) subwords[8]

---

[8] We use the same BPE tokenization and vocabulary as Ghazvininejad et al. (2019).

as the joint source-target dictionary. We evaluate performance with BLEU (Papineni et al., 2002) for both language pairs[9].

For inference efficiency, we report decoding speedup over beam search. Specifically, we test the inference speed by running the model with one sentence at a time (batch=1). We perform model inference with fairseq implementation[10] using Pytorch 1.10.1 with 1 Nvidia Tesla P100-PCIe of 16GB GPU memory under CUDA 11.1.

**Model Configuration** The primary target model we accelerate in our experiments is the Transformer-base model with a 6-layer encoder and a 6-layer decoder of 512/2048 embedding/FFN dimension, which can achieve state-of-the-art results on the benchmarks under comparable model size conditions. For the Spec-Drafter, we adopt a similar architecture to the AR model except with 12 encoder layers and 2 decoder layers to make sure it adheres to both the Capability and Latency principles. We apply sequence-level knowledge distillation (Kim and Rush, 2016) by the AR teacher to the Spec-Drafter to align its behavior with the AR model as much as possible. We include model training details in Appendix A. For the Spec-Verification, we find the hyperparameters $\beta$ and $\tau$ leading to the best generation quality on the validation set. Besides, we re-implement Blockwise Decoding[11] using the same device and environment as ours to facilitate fair comparison.

### 4.2 Results

We present the performance and the acceleration effect of SpecDec to Transformer in Table 1. As reported in the previous work (Stern et al., 2018), Blockwise Decoding ($k = 10$) can only achieve $1.4\times \sim 2\times$ speedup without affecting the generation results over the Transformer-base model. Further increasing the parallel capability of Blockwise Decoding (e.g., $k = 25$) will not introduce more speedup as its limited drafting accuracy prevents more drafted tokens from being accepted. In contrast, our SpecDec shows consistent performance improvement with increased parallel capabilities ($k = 10 \rightarrow k = 25$), resulting in around

---

[9] We also report sacreBLEU (Post, 2018) and COMET (Rei et al., 2020) scores in Appendix C.

[10] https://github.com/pytorch/fairseq

[11] In the original paper of Blockwise Decoding, there is also a variation that allows the AR model to be fine-tuned for better drafting tokens. We don't discuss this variation because it severely affects the generation quality.

| Models | EN→DE | | DE→EN | | EN→RO | | RO→EN | |
|---|---|---|---|---|---|---|---|---|
| | **Speed** | **BLEU** | **Speed** | **BLEU** | **Speed** | **BLEU** | **Speed** | **BLEU** |
| Transformer-base ($b = 5$) | 1.0× | 28.89 | 1.0× | 32.53 | 1.0× | 34.96 | 1.0× | 34.86 |
| Transformer-base ($b = 1$) | 1.1× | 28.73 | 1.1× | 32.18 | 1.1× | 34.83 | 1.1× | 34.65 |
| Blockwise Decoding ($k = 10$) | 1.9× | 28.73 | 2.0× | 32.18 | 1.4× | 34.83 | 1.4× | 34.65 |
| Blockwise Decoding ($k = 25$) | 1.6× | 28.73 | 1.7× | 32.18 | 1.2× | 34.83 | 1.2× | 34.65 |
| SpecDec ($k = 10$) | 4.2× | 28.90 | 4.6× | **32.61** | 3.9× | 35.29 | 4.1× | 34.88 |
| SpecDec ($k = 25$) | **5.1×** | **28.93** | **5.5×** | 32.55 | **4.6×** | **35.45** | **4.8×** | **35.03** |
| 12+2 Transformer-base ($b = 5$) | 1.0× | **29.13** | 1.0× | 32.45 | 1.0× | 34.93 | 1.0× | 34.80 |
| 12+2 Transformer-base ($b = 1$) | 1.1× | 28.99 | 1.1× | 32.08 | 1.1× | 34.79 | 1.1× | 34.55 |
| Blockwise Decoding ($k = 10$) | 1.6× | 28.99 | 1.7× | 32.08 | 1.2× | 34.79 | 1.2× | 34.55 |
| Blockwise Decoding ($k = 25$) | 1.4× | 28.99 | 1.5× | 32.08 | 1.1× | 34.79 | 1.1× | 34.55 |
| SpecDec ($k = 10$) | 2.7× | 29.08 | 3.0× | 32.40 | 2.3× | **35.12** | 2.4× | 34.85 |
| SpecDec ($k = 25$) | **3.0×** | **29.13** | **3.3×** | 32.48 | **2.5×** | 35.07 | **2.6×** | **34.91** |

Table 1: The performance of Speculative Decoding (SpecDec) to speed up the Transformer-base on the WMT benchmarks. We re-implement and evaluate Blockwise Decoding using the same device and environment as ours.

| Models | Tok. | BLEU | $t_d$ | Speed |
|---|---|---|---|---|
| Transformer-base ($b = 5$) | 1.00 | 28.89 | - | 1.0× |
| SpecDec | | | | |
| └ *w/* head-based drafter | 2.32 | 28.02 | 0.81 | 1.7× |
| └ *w/* Spec-Drafter (w/o Principle I) | 7.05 | 28.56 | 2.29 | 4.4× |
| └ *w/* Spec-Drafter (w/o Principle II) | 8.21 | 28.95 | 10.87 | 4.0× |
| └ *w/* Spec-Drafter | **8.23** | 28.93 | 5.21 | **5.1×** |

Table 2: Ablation studies of the drafter in the SpecDec on WMT14 EN→DE. **Tok.** denotes the average number of drafted tokens accepted in each iteration. $t_d$ denotes the average time cost of drafting per iteration. The head-based drafter is the one used by Blockwise Decoding (Stern et al., 2018). The Spec-Drafter (w/o Principle I) reduces the model/FFN dimension to 256/1024, while the Spec-Drafter (w/o Principle II) does not use a deep encoder and shallow decoder but instead utilizes a more balanced architecture with equal depth of encoder and decoder layers.

4.6×∼5.5× speedup across the translation benchmarks; moreover, it even achieves an improvement in generation quality (by the BLEU metric) compared with AR greedy decoding.

Similar results are also observed when accelerating the Transformer with a 12-layer encoder and 2-layer decoder – SpecDec can still achieve around 2.5×∼3.3× speedup while Blockwise Decoding's acceleration effect becomes nearly negligible over the fast AR baseline.

### 4.3 Analysis

In this section, we conduct a comprehensive and thorough analysis, to demonstrate that the significant improvement of the SpecDec arises from both the Spec-Drafter (Section 4.3.1) and Spec-Verification (Section 4.3.2).

#### 4.3.1 Drafting

According to Table 2, the Spec-Drafter significantly outperforms the head-based drafter (as used in Blockwise Decoding) in terms of both end-to-

| Models | $k$ | Tok. | BLEU | Speed |
|---|---|---|---|---|
| AR-base ($b = 5$) | - | 1.00 | 26.72 | 1.00× |
| SpecDec | 10 | 6.05 | 26.80 | 3.99× |
| | 15 | 6.93 | **27.12** | 4.54× |
| | 20 | 7.41 | 27.05 | 4.72× |
| | 25 | **7.89** | 26.97 | **5.04×** |
| | 30 | 7.67 | 26.89 | 4.82× |

Table 3: The mean accepted tokens (**Tok.**), the generation quality (*BLEU*), and the efficiency (*Speed*) when decoding with a various number of block size $k$ on the development set of WMT14 EN→DE.

end generation quality and efficiency. To further investigate the Spec-Drafter, we conduct an ablation study on the principles it follows. Ablating the Capability Principle by reducing its size results in a drastic drop in end-to-end acceleration performance, as more iterations are needed to complete the decoding process, indicated by a lower **Tok.**. When we ablate the Latency Principle by using a balanced (6+6) encoder-decoder architecture for

| Models | $\tau$ | Top-3 ($\beta = 3$) | Top-5 ($\beta = 5$) |
|---|---|---|---|
| Vanilla | - | 6.41/26.62 | |
| SpecDec | 1 | 7.89/**26.97** (5.0×) | 7.92/26.88 |
| | 2 | 8.75/26.84 | 8.83/26.79 |
| | 3 | 9.51/26.71 | 9.64/26.68 |
| | 4 | 10.11/26.60 | 10.63/26.59 |
| | 5 | 10.46/26.58 | **11.01**/26.58 (6.8×) |

Table 4: Results of SpecDec ($k = 25$) on the development set of WMT14 EN→DE with different hyperparameters. Each cell lists the mean accepted tokens and BLEU score. Among the runs in the table, the highest BLEU score of 26.97 is achieved when $\beta = 3$ and $\tau = 1.0$, with a 5× speedup. On the other hand, when $\beta = 5$ and $\tau = 5$, the highest **Tok.** (i.e., 11.01) is reached, resulting in almost 7× speedup, though the BLEU score slightly decreases to 26.58.

the drafter, it also experiences a substantial decline in end-to-end acceleration performance due to increased latency in each iteration (reflected by a higher $t_d$).

Moreover, we analyze the block size $k$'s effect on the end-to-end acceleration performance of the paradigm in Table 3. In contrast to the blockwise decoding achieving its best acceleration performance at $k = 10$ as shown in Table 1, SpecDec achieves its best performance at $k = 25$ with 7.89 mean accepted tokens each iteration. Further increasing $k$ has an adverse effect, because it will become very hard for the model to learn to draft too many tokens simultaneously given the model capacity, resulting in a drop of **Tok.**.

### 4.3.2 Verification

We study the effect of Spec-Verification on the development set (i.e., *newstest-2013*) of WMT14 EN→DE in Table 4. Moderately increasing $\tau$ and $\beta$ in the Spec-Verification not only leads to an increase of mean accepted tokens (**Tok.**) and speed since AR verification becomes less strict but also improves the generation quality over greedy decoding. However, the generation quality may decrease if over-relaxed: the BLEU score will degrade from the peak of 26.97 to 26.58 when decoding with *top-5* selection (i.e., $\beta = 5$) and $\tau = 5.0$. Based on the results in the development set, we select $\beta = 3, \tau = 1.0$ as our Spec-Verification hyperparameters.

### 4.4 Practical Value

In addition to the remarkable speedup results, we demonstrate SpecDec's additional advantages that

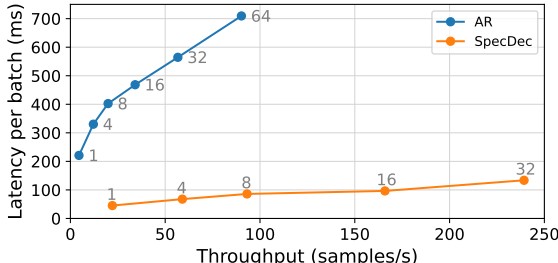

Figure 5: The latency-throughput curve with various batch sizes on WMT14 EN→DE.

enhance its practical value in the following three aspects:

**Better latency-throughput trade-off** SpecDec achieves inference acceleration by increasing the GPU computing parallelism. Although increasing the batch size can also increase the computing parallelism to improve throughput, it results in increased latency, which is not desirable in real-world application scenarios. Therefore, a smaller batch size is often employed during inference, but this in turn results in the underutilization of GPU computing resources, leading to the dilemma of low throughput for small batches and high latency for large batches, as illustrated by Figure 5. SpecDec effectively addresses this dilemma. Even by maintaining a small batch size, SpecDec can fully utilize the computing performance of the GPU, significantly improving both efficiency and throughput.

**Easily adaptable for existing models** In many practical applications, generative models are often pretrained with massive data, which exhibits very high performance. Developing a faster model from scratch to replace the pretrained model is highly challenging and typically requires substantial computational costs to reiterate the pretraining process. Otherwise, the quality of the new models is very likely to be compromised despite the increased speed, as Table 5 shows. However, SpecDec can be easily adapted to accelerate existing pretrained models. Taking the BART-base (Lewis et al., 2020) model for the abstractive summarization task as an example, we can easily achieve 5.1× speedup without compromising the generation quality only by initializing the Spec-Drafter with the BART-base encoder and training it with the BART-base distilled summarization training set.

**Retaining the behavior of the original model** As introduced in Section 1, one significant advantage of SpecDec is that it does not develop a new

| | Models | Rouge-1 | Rouge-2 | Rouge-L | Speed |
|---|---|---|---|---|---|
| AR | BART-base ($b = 5$) | 43.08 | 20.41 | 40.15 | 1.0× |
| | BART-base ($b = 1$) | 43.00 | 20.28 | 39.96 | 1.1× |
| NAR | GLAT+CTC (Qian et al., 2021) | 37.76 ↓5.24 | 14.08 ↓6.20 | 33.69 ↓6.27 | 14.5× |
| | DAT (Huang et al., 2022) | 38.95 ↓4.05 | 16.11 ↓4.17 | 35.43 ↓4.53 | 14.1× |
| | CMLM (Ghazvininejad et al., 2019) | 37.59 ↓5.41 | 15.17 ↓5.11 | 34.22 ↓5.54 | 1.8× |
| | RewriteNAT (Geng et al., 2021) | 39.12 ↓3.88 | 16.24 ↓4.04 | 35.74 ↓4.22 | 3.1× |
| SpecDec | SpecDec ($k = 25$) | 43.11 ↑0.11 | 20.43 ↑0.15 | 40.19 ↑0.23 | **5.1×** |

Table 5: Results of different methods for Abstractive Summarization on CNN-DM (Hermann et al., 2015). The results on par with BART-base performance are highlighted in orange, while blue denotes performance degradation. Compared to prior NAR methods, SpecDec can be easily adapted to accelerate the BART model only by downstream task fine-tuning.

| Models | BLEU |
|---|---|
| Transformer-base (greedy) | 100.00 |
| GLAT+CTC (Qian et al., 2021) | 59.10 |
| DAT (Huang et al., 2022) | 63.79 |
| CMLM (Ghazvininejad et al., 2019) | 60.15 |
| RewriteNAT (Geng et al., 2021) | 65.42 |
| Deep-Shallow (Kasai et al., 2020) | 64.66 |
| SpecDec ($k = 25$) | **86.52** |

Table 6: Relative BLEU score computed between the generation of the *existing* Transformer (i.e., the target model) and other models/approaches. SpecDec shows much better alignment with the target model's behavior than others.

faster model to replace the existing model. Instead, it accelerates the existing model with minimal changes to its behavior. As shown in Table 6, the consistency (BLEU) of SpecDec's generated results with the original model exceeds 85%, while that of a newly built fast NAR model is only around 55%. The characteristic of maintaining the behavior of the original model makes SpecDec even more valuable in practical applications because transitioning from a well-tested and mature model to one with substantially different behavior is risky, requiring extensive recalibration and various offline evaluations and online feedback in practice.

## 5    Related Work

**Speculative Decoding**    We have demonstrated since early 2022 (see our arXiv preprints in 2022) that our proposed methodology, which formally introduces an independent model as a drafter combined with an advanced verification strategy to fully exploit speculative execution, is promising and has potential to evolve into a *de facto* standard in the future for efficient and lossless de-

coding. Since this work was proposed, we are pleased to see an increasing number of following studies (Leviathan et al., 2023; Chen et al., 2023; Kim et al., 2023; Spector and Re, 2023; Zhang et al., 2023) acknowledge, explore and adopt this methodology to accelerate Transformer inference. Among them, Leviathan et al. (2023) use the same name as ours (i.e., Speculative Decoding), employing a small AR model as a drafter[12] as well as advanced sampling algorithm. Chen et al. (2023) is similar to Leviathan et al. (2023) but it was the first to validate this methodology to accelerate a large language model (i.e., 70B Chinchilla) with a 4B drafter model, thus receiving the most attention. SpecInfer (Miao et al., 2023) proposed to utilize various boost-tuned small language models for joint drafting, to improve the speculation accuracy of the LLM's outputs. Besides, it introduces an advanced token tree verification strategy to verify all candidate token sequences in parallel. Distill-Spec (Zhou et al., 2023) further investigated the efficacy of knowledge distillation in enhancing the alignment between the target model and the drafter in speculative decoding. In addition to employing additional models as drafters, there has also been some research that proposes various strategies to efficiently generate drafts from the LLM itself (Santilli et al., 2023; Zhang et al., 2023). All the following research strongly backs up the value of this original work.

**Early *Draft-then-verify* attempts**    This work is a generalized version of our previously proposed (Input-guided) Aggressive Decoding[13] (Sun et al.,

---

[12]We provide a detailed comparison between Leviathan et al. (2023) and our work in Appendix G.

[13]As our technical report (Ge et al., 2022b) in May 2022 discusses, the Input-guided Aggressive Decoding is indeed a special case of Speculative Decoding.

2021) in Grammatical Error Correction (GEC), which assumes that the input is exactly the sentence to be generated in the future and then verifies the whole sentence in parallel. Blockwise Decoding (Stern et al., 2018) inserted $k - 1$ feedforward heads on top of the Transformer decoder to generate $k$ positions in parallel and used the original head to verify these outputs. However, both the above studies did not fully investigate the potential of this paradigm and thus failed to uncover its great value for efficient seq2seq generation: Sun et al. (2021) only works for tasks whose inputs and outputs are highly similar (e.g., GEC). Stern et al. (2018) overlooked the importance of drafting accuracy; as a result, their underinvested prediction heads severely limit the acceleration results. In contrast, we conduct thorough investigations and fully exploit speculative execution, refreshing the impression of its limited acceleration potential and revealing its real value in practice.

**Non-autoregressive Decoding**   There is also another line of work named Non-Autoregressive Decoding (NAR) (Gu et al., 2018), which decodes multiple tokens in parallel compared with conventional AR, thus showing remarkable superiority in inference efficiency. Recently, various attempts have been made to improve the performance of NAR models, including training with alignment-based objectives (Libovický and Helcl, 2018; Ghazvininejad et al., 2020; Saharia et al., 2020; Gu and Kong, 2021; Shao and Feng, 2022), modeling dependencies between target tokens (Ghazvininejad et al., 2019; Shu et al., 2020; Qian et al., 2021; Bao et al., 2021) and designing various model architectures (Zheng et al., 2021; Huang et al., 2022). As discussed in Section 4.4, replacing a powerful pretrained model with NAR models in practice is challenging due to the substantial computational costs required to reiterate the pretraining process. Additionally, transitioning from a well-tested and mature model to a new NAR model with significantly different behavior poses risks in practical applications. In contrast, our proposed SpecDec can be conveniently adapted to speed up *existing* AR models including high-performance pretrained models like BART with little effort. Moreover, SpecDec minimally alters the behavior of *existing* models, showcasing its ability to preserve reliable generation performance in real-world practical applications.

## 6   Conclusion

We present Speculative Decoding (SpecDec), the first work to explicitly embrace the idea of speculative execution for seq2seq generation acceleration with a formal study and extensive discussion of both drafting and verification phases. Contrary to the common belief that an increase in model complexity tends to hamper inference speed, SpecDec's introduction of an appropriately invested auxiliary drafter model substantially speeds up Transformer inference, owing to higher computational parallelism introduced by speculative execution to better utilize computing resources.

The remarkable acceleration performance, combined with the advantages demonstrated in our experiments, clearly illustrates that SpecDec is a practical acceleration method for model deployment in real-world applications. We hope that our preliminary study could draw more attention to this promising decoding paradigm that may potentially evolve into a *de facto* standard for efficient Transformer decoding in the near future.

## Limitations

Compared with conventional autoregressive decoding, SpecDec introduces an extra Spec-Drafter module for ensuring its drafting accuracy, which brings additional memory cost at test time. Therefore, SpecDec is particularly suitable for inference scenarios where GPU memory is abundant but there is an urgent need to improve latency – it provides a solution to trading the surplus GPU memory for speed improvements. As thoroughly discussed in Appendix B, such scenarios are very common in practice. Most importantly, memory is no longer the bottleneck for practical model deployment. With the emergence and maturity of various data/tensor/pipeline parallelism techniques, the addition of more GPUs can easily address memory issues, which is also why models continue to grow larger. In contrast, latency remains an inescapable bottleneck in model deployment that cannot be resolved merely by increasing the number of machines. Therefore, we believe the increased memory consumption may not severely affect its practical value.

## Acknowledgements

We thank Fan Yang, Lingxiao Ma, and Lidong Zhou in Microsoft Research for their constructive

comments on this work. This paper is supported by the National Key Research and Development Program of China 2020AAA0106700 and NSFC project U19A2065.

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

| Hyperparameter | Value |
|---|---|
| devices | 8 Nvidia V100 GPU |
| label smoothing | 0.1 |
| # max tokens | 4096 |
| update frequency | 4 |
| dropout rate | [0.1, 0.2, 0.3] |
| max source positions | 1000 |
| max target positions | 1000 |
| Adam lr | $5 \times 10^{-4}$ |
| Adam $\beta_1$ | 0.9 |
| Adam $\beta_2$ | 0.999 |
| Adam $\epsilon$ | $1 \times 10^{-6}$ |
| lr-scheduler | inverse square |
| warm-up lr | $1 \times 10^{-7}$ |
| weight decay | 0.01 |
| clip norm | 5.0 |
| # warmup updates | 10000 |
| max updates | 300K |

Table 7: Hyper-parameters and settings of Spec-Drafter.

# Appendix

## A    Hyperparameters

Hyper-parameters of training our proposed Spec-Drafter are listed in Table 7. Following Vaswani et al. (2017) and Ott et al. (2018), we also average model parameters from the last 10 checkpoints.

## B    Memory Analysis

### B.1    Additional Memory Cost by SpecDec

The peak memory footnote of SpecDec during inference mainly comes from three parts:

- Static AR model's weights
- Static Spec-Drafter's weights
- Intermediate variables/results

Compared with AR, the additional memory cost of SpecDec comes from the last two parts. While the static Spec-Drafter's weights account for the majority of the additional memory cost, the additional cost for storing intermediate variables is negligible because the Spec-Drafter and AR model decode alternatively during inference. Compared with AR, SpecDec's additional intermediate variables/results include:

- The Spec-Drafter's last encoder layer's representation that will not be freed until decoding finishes, which is equal to $B \cdot S \cdot d$ where $B$ is the batch size, $S$ is the sequence length and $d$ is the dimension of the model. This part is actually negligible: for example, when $B = 32$, $S = 128$, $d = 512$, this part's memory cost is only 8MB (fp32) / 4MB (fp16).

- The largest intermediate variables/results during inference:
  - For a short sequence (e.g., sentence-level inputs/outputs in MT tasks), the largest intermediate variable is the output tensor after the Spec-Drafter's/AR model's vocabulary projection layer − $B \cdot |V| \cdot k$ where $B$ is the batch size, $|V|$ is the vocabulary size and $k$ is the block size. Compared with the memory cost for storing the Spec-Drafter's weights, this part is usually smaller. Also $B \cdot k$ tokens can be easily divided into small batches (e.g., –softmax-batch in fairseq) for vocabulary projection to avoid massive memory cost in case $B \cdot |V| \cdot k$ is large.
  - For a long sequence (e.g., paragraph-level inputs/outputs in summarization tasks), the largest intermediate variable becomes the tensor for storing self-attention computation whose size increases quadratically with $S$ ($S$ is the sequence length). This variable accounts for the largest memory cost for storing intermediate results in both AR and SpecDec. Therefore, in this case, this part does not introduce additional memory cost compared with AR.

Table 8 and Table 9 show the comparisons of peak GPU memory footprint[14] (MB) between SpecDec and AR (during inference) on the above two scenarios (i.e., MT and summarization). The results are consistent with our analysis above:

> **The majority of the additional memory cost (i.e., $\Delta$Memory) is for storing the Spec-Drafter's weights and the additional memory cost is not very likely to significantly increase as the batch size or sequence length increases.**

Our experiments above pre-loaded both the Spec-Drafter and AR model. In fact, it is also possible to load the static weights of the AR model and Spec-Drafter in a lazy loading manner in the meantime of GPU computation to save memory as they run alternatively. However, it is usually unnecessary in practice, because for a seq2seq model deployed on modern GPUs for online service, **it is latency rather than memory that is the performance bottleneck**. See the next section for more discussion.

---

[14]Tested with `torch.cuda.max_memory_allocated()`

| Models | Model Weights | Batch Size | | | | |
|---|---|---|---|---|---|---|
| | | 1 | 4 | 8 | 16 | 32 |
| AR (greedy) | 232.4 | 243.2 | 271.7 | 301.4 | 366.4 | 494.6 |
| SpecDec ($k = 25$) | 477.8 | 491.5 | 519.8 | 559.2 | 634.1 | 782.5 |
| $\Delta$Memory | 245.4 | 248.3 | 248.1 | 247.8 | 267.7 | 287.9 |

Table 8: Peak GPU memory utilization on WMT14 EN-DE translation dataset. The results are obtained with fp32 on a single Nvidia P100 GPU.

| Models | Model Weights | Memory Cost |
|---|---|---|
| AR (greedy) | 534.6 | 696.9 |
| SpecDec | 1089.6 | 1264.6 |
| $\Delta$Memory | 555.0 | 567.7 |

Table 9: Peak GPU memory utilization on CNN-DM with batch size 1 with fp32 on a single Nvidia P100 GPU.

## B.2 Memory Is Rarely the Bottleneck

To understand the performance bottleneck of online deployed seq2seq models, we test the latency and memory cost of T5-large[15] (around 770M parameters) with fp16 on 1 Nvidia A40 GPU running greedy decoding in the machine translation and abstractive summarization task, and show results in Table 10 and 11.

| Statistics | Batch Size | |
|---|---|---|
| | 1 | 32 |
| Latency (s) | 1.0▲ | 1.4▲ |
| Memory Util. (MB) | 1482 | 2003 |
| Memory Util. (%) | 3.0 | 4.0 |

Table 10: Latency and peak GPU memory utilization of T5-Large on WMT14 EN-DE.

| Statistics | Batch Size | |
|---|---|---|
| | 1 | 32 |
| Latency (s) | 2.7▲ | 4.7▲ |
| Memory Util. (MB) | 2999 | 7230 |
| Memory Util. (%) | 6.2 | 15.0 |

Table 11: Latency and peak GPU memory utilization of T5-Large on CNN-DM.

For MT, T5-large's latency is over 1 second which is actually too long to be accepted because most MT engines in practice require the latency to be less than 100ms. However, its memory cost is

only less than 2GB – far below A40 GPU's memory capacity (i.e., 48GB[16]).

For abstractive summarization, even if the batch size increases to 32, its memory cost is still less than 50% utilization of 1 A40 GPU but its latency is already close up to 5 seconds that is too long for an online service in practice.

To sum up, we now understand latency is the bottleneck of seq2seq models for online deployment in most cases. Therefore, we do not think additional memory cost by SpecDec will undermine its practical value; instead, we think a significant lossless acceleration even at the cost of memory (i.e., time–memory trade-off) is much more meaningful than the acceleration at the cost of quality, which should be the right path that we need to pay more attention to given much memory headroom on modern GPUs.

| Models | BLEU | SacreBLEU | COMET |
|---|---|---|---|
| AR-base ($b = 5$) | 28.89 | 28.2 | 51.90 |
| AR-base ($b = 1$) | 28.73 | 28.0 | 51.53 |
| SpecDec ($k = 25$) | **28.93** | **28.2** | **52.10** |

Table 12: SacreBLEU and COMET scores on WMT14 EN-DE.

## C  SacreBLEU and COMET Scores

Despite tokenized BLEU scores, we also report SacreBLEU[17] (Post, 2018) and COMET[18] (Rei et al., 2020) scores in Table 12 and 13 to provide a reference for future research. SpecDec can also achieve performances on par with the AR model with the evaluation in sacreBLEU and COMET. Schmidt et al. (2022) pointed out that inconsistencies in the use of tokenized BLEU lead to deviations of up to 1.8 BLEU points. Therefore, we

---

[15]In practice, T5-large is rarely deployed for online service because it is too large and expensive to serve.

[16]It can also easily scale to 96GB or more with NVIDIA NVLink connection of multiple GPUs or multi-node connection.

[17]https://github.com/mjpost/sacrebleu

[18]Obtained with `wmt20-comet-da` from version 1.1.0.

| Models | EN→DE | DE→EN | EN→RO | RO→EN |
|---|---|---|---|---|
| Transformer-base ($b = 5$) | 28.89(28.2[†]) | 32.53(32.1[†]) | 34.96(34.0[†]) | 34.86(34.2[†]) |
| Transformer-base ($b = 1$) | 28.73(28.0[†]) | 32.18(31.7[†]) | 34.83(33.9[†]) | 34.65(33.9[†]) |
| SpecDec ($k = 25$) | **28.93(28.2[†])** | **32.55(32.1[†])** | **35.45(34.5[†])** | **35.03(34.3[†])** |

Table 13: BLEU and SacreBLEU (denoted by [†]) scores on WMT14 EN-DE and WMT16 EN-RO benchmarks.

| Decoding Algorithm | BLEU | Speed |
|---|---|---|
| AR (greedy) | 32.23 | 1.0× |
| Leviathan et al. (2023) (0.1B drafter, T5-Small) | 32.23 | 2.8× |
| Leviathan et al. (2023) (0.8B drafter, T5-Large) | 32.23 | 1.6× |
| SpecDec ($k = 25$) (0.5B drafter) | **32.31** | **4.9×** |

Table 14: Comparison of SpecDec and Leviathan et al. (2023) in accelerating T5-XXL (11B) for WMT14 EN-DE translation.

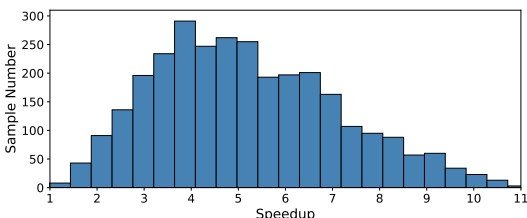

Figure 6: Single sentence speedup distribution by SpecDec ($k = 25$) compared with Transformer-base ($b = 5$). The results are obtained with WMT14 EN→DE.

recommend that future research use sacreBLEU when comparing with our work.

## D Speedup Distribution

Figure 6 presents SpecDec's speedup distribution of a single sentence on the WMT14 EN→DE test set (which has 3,003 sentences in total), showing that most sentences are translated with a 3×∼7× speedup compared to AR beam search, while some rare cases can even achieve over 10×∼11× speedup.

## E Discussions of Beam Search

For possible concerns that SpecDec may not apply beam search, we make three points here:

1. As Kim and Rush (2016) mentioned, knowledge distillation largely decreases the performance gap of beam search and greedy decoding. In practice, greedy decoding can actually be comparable to beam search results after KD.
2. In practical online deployment, KD is almost used by default for enhancing the results for student models and greedy decoding is much more common than beam search because it is more cost-effective – it not only runs faster than beam search but also achieves decent performance with a student model trained through KD (as Point 1 addressed)
3. Beam search is also an approximate and heuristic solution, which is not a golden rule. In fact, Spec-Verification works in a similar way as beam search – it is also an approximate and heuristic solution by considering n-best and scores, which can be considered as an approximation of beam search. As shown in Table 1, it achieves comparable performance to beam search but much faster (4× ∼ 6×).

## F Carbon Emission

SpecDec introduces extra computational overhead, which leads to an increase in GPU power consumption. However, it can substantially reduce GPU hours owing to its high efficiency. We compare GPU power consumption and GPU hours of autoregressive decoding (AR) and SpecDec for translating 3000 sentences in Table 15.

We follow a formula for Wu et al. (2022) to calculate the total energy consumption and carbon emitted: $E = P \times t \times$ PUE, where we set PUE (Power Usage Effectiveness) at 1.1. $CO_2eq = 0.385g \times CO_2eq/Wh \times E$, where $0.385g*CO_2eq/Wh$ is the carbon intensity factor that is set based on the US national average.

According to Table 15, while SpecDec's GPU power consumption is 28% higher, its GPU hours are 540% shorter than autoregressive decoding. As a result, SpecDec's total energy consumption

| Models | GPU Type | GPU Avg. Power consumption (P) | GPU-hours (t) | Total Energy consumption (E) | Carbon emitted (CO$_2$eq) |
|---|---|---|---|---|---|
| AR (greedy) | Nvidia P100 | 86W | 0.27h | 25.54Wh | 9.83g |
| SpecDec ($k = 25$) | Nvidia P100 | 110W | 0.05h | 6.05Wh | 2.33g |

Table 15: Carbon footprint of autoregressive decoding and SpecDec using the same P100 GPU device. We follow Wu et al. (2022) to compute the carbon emission of the two decoding strategies under the same device. The results were obtained on WMT14 EN-DE.

| SOURCE | According to the details provided , the tunnel had not yet been put into use . |
|---|---|
| *Draft* | Nach den Angaben Angaben war der Tunnel noch nicht in |
| *Verify* | Nach den vorliegenden ~~war war der Tunnel noch nicht in Betrieb~~ |
| *Draft* | Nach den vorliegenden Angaben war der Tunnel noch nicht in Betrieb genommen `[EOS]` |
| *Verify* | Nach den vorliegenden Angaben war der Tunnel noch nicht in Betrieb genommen worden |
| *Draft* | Nach den vorliegenden Angaben war der Tunnel noch nicht in Betrieb genommen worden . `[EOS]` |
| *Verify* | Nach den vorliegenden Angaben war der Tunnel noch nicht in Betrieb genommen worden . `[EOS]` |
| RESULTS | Nach den vorliegenden Angaben war der Tunnel noch nicht in Betrieb genommen worden . |

Table 16: Examples from the WMT14 English-German translation task. At each iteration, *Draft* and *Verify* are the outputs of the Spec-Drafter and the AR verification, respectively. Tokens within red blocks are the bifurcation positions. The verification pieces after the bifurcation are annotated as strikethrough. The highlighted parts are translations of previous iterations. The hyperparameters are $k = 10$, *top-3*, $\tau = 1.0$. The output pieces after the `[EOS]` token is omitted in the table.

is 23.7% of autoregressive decoding, resulting in 4.2× reduction of carbon emission.

Therefore, SpecDec not only does not increase carbon emission but actually significantly reduces it, making it a more environmentally friendly option for inference.

## G  Comparison with Other Work

### G.1  Speculative Decoding with an Autoregressive Drafter

Following this work, some subsequent research (Leviathan et al., 2023; Chen et al., 2023) has also explored using AR models (e.g., smaller language models) as independent drafters to accelerate inference. However, for seq2seq generation, the acceleration effect of AR drafting is severely limited, resulting in an end-to-end speedup at 1.6×∼2.8×, which is far lower than our 4.9× speedup.

Table 14 illustrates the detailed comparison of our re-implemented Leviathan et al. (2023) and ours in accelerating T5-XXL (11B) for WMT14 EN-DE translation. We implemented our Spec-Drafter with a 24-layer encoder and a 6-layer decoder for comparison. Its embedding/FFN dimension/#heads are 1024/4096/16. As shown in Table 14, our approach uses a 0.5B Spec-Drafter to achieve an almost 5× speedup without qual-

ity degradation, while Leviathan et al. (2023)'s speedup results are lower than 3×. The reasons for our approach significantly outperforming Leviathan et al. (2023) are twofold:

- Our Spec-Drafter is specially learned to draft for accelerating the target model with the speculation execution idea: its draft results are better aligned with the target model's results.

- Our approach adopts a fast deep-encoder-shallow-decoder architecture as well as a non-autoregressive approach to generate drafts, which is significantly more efficient than their autoregressive drafting method.

### G.2  Speculative Decoding in Special Cases

Our previous paper (Sun et al., 2021) and technical reports (Ge et al., 2022b; Yang et al., 2023) present extensive empirical studies of Speculative Decoding in special cases (e.g., text editing or retrieval-augmented generation) and compare it with competitive approaches (Malmi et al., 2019; Omelianchuk et al., 2020; Chen et al., 2020) in those scenarios. We recommend interested readers to refer to our previous work for further insights into these specific applications.

## H Case Study

In Table 16, we represent an example to illustrate how SpecDec generates translations. In the first iteration, the outputs of the Spec-Drafter are non-autoregressive with *multi-modality* problems like "Angaben Angaben". The verifier accepts tokens of "Nach den" and replaces the inappropriate translation "Angaben" with "vorliegenden". All the drafted tokens after the bifurcation position (i.e. marked as red tokens in Table 16) are all discarded. In the second iteration, Spec-Verification finds the bifurcation at the last position, thus all tokens before this position are accepted. After 3 iterations, the decoding is finished since the [EOS] token is found.