# OpenReview forum: "Speculative Decoding: Exploiting Speculative Execution for Accelerating Seq2seq Generation"
_EMNLP/2023/Conference — EMNLP 2023 Findings_

### Official Review · Reviewer_PLjh · 2023-08-03

**Soundness:** 2

**Excitement:**

3: Ambivalent: It has merits (e.g., it reports state-of-the-art results, the idea is nice), but there are key weaknesses (e.g., it describes incremental work), and it can significantly benefit from another round of revision. However, I won't object to accepting it if my co-reviewers champion it.

**Paper Topic And Main Contributions:**

This paper proposes Speculative Decoding to speed up autoregressive decoding in seq2seq generation. The key contributions include Spec-Drafter -- a novel model specially optimized for drafting -- and Spec-Verification -- an advanced method for verification in the decoding paradigm. Authors conduct experiments on machine translation and abstractive summarization and show the proposed SpecDec is more efficient than baselines.

**Reasons To Accept:**

1) Clear motivation and writing. Accelerating AR is a very important research topic in the sequence generation community.
2) Strong results on machine translation and abstractive summarization show the effectiveness of the approach used.
3) The two modifications Spec-Drafter and Spec-Verification are interesting and smartly designed.

**Reasons To Reject:**

1) The proposed approach consumes more memory during decoding.
2) I cannot agree that memory is rarely the bottleneck. It is indeed okay in this paper because of the setup. However, for larger models and longer sequences (which are exactly the current large transformer use case), why memory is not a problem?
3) It seems that the proposed approach introduces more computational overhead during inference. Is there any comparison about Carbon Emission? My concern is, even if the approach can generate faster, it actually consumes much more energy than before. I may save some money but actually, it is not that good for the environment.

**Reproducibility:**

3: Could reproduce the results with some difficulty. The settings of parameters are underspecified or subjectively determined; the training/evaluation data are not widely available.

**Reviewer Confidence:**

3: Pretty sure, but there's a chance I missed something. Although I have a good feel for this area in general, I did not carefully check the paper's details, e.g., the math, experimental design, or novelty.

---

> ### Author Rebuttal · Authors · 2023-08-27
>
> Thank you for your valuable review comments. However, there is a serious **misunderstanding regarding the memory cost issue** and other concerns you raised. We hope our explanation and responses can resolve your misunderstanding and concerns, and improve your impression and evaluation of our paper:
>
> &nbsp;
>
> ***Q1: The proposed approach consumes more memory during decoding. I cannot agree that memory is rarely the bottleneck. It is indeed okay in this paper because of the setup. However, for larger models and longer sequences (which are exactly the current large transformer use case), why memory is not a problem?***
>
> R1: We have acknowledged in the limitations section that SpecDec introduces memory overhead as a trade-off for accelerating decoding. However, the benefits of our approach, including significant inference speedup and substantial improvements in the latency-throughput curve, **far outweigh the memory overhead**. As extensively discussed in Appendix B, the memory overhead introduced by SpecDec primarily stems from storing the weights of the draft model. Generally, this additional memory consumption is not excessive and does not increase significantly with the sequence length or batch size. Most importantly, memory is no longer the bottleneck for practical model deployment. With the emergence and maturity of **various data/tensor/pipeline parallelism techniques**, the addition of more GPUs **can easily address memory issues**, which is also why models continue to grow larger. In contrast, latency remains an inescapable bottleneck in model deployment that cannot be resolved merely by increasing the number of machines.
>
> **Precisely because memory is no longer the primary bottleneck in practical model deployment**, and the **urgency of addressing latency** issues has increased, there has been a growing interest in research **following our work to explore the methodology** of introducing a draft model to improve transformer inference. As **the first work** to propose the name "Speculative Decoding" and explicitly leverage the idea of speculative execution (You can consulate with SAC/AC to confirm this fact), **our approach has been acknowledged** by several recent papers studying this methodology:
>
> 1. Accelerating Large Language Model Decoding with **Speculative** Sampling. (Deepmind)
> 2. SpecTr: Fast **Speculative** Decoding via Optimal Transport. (Google Research)
> 3. SpecInfer: Accelerating Generative LLM Serving with **Speculative** Inference and Token Tree Verification
>
> **All these papers follow our work** to introduce a drafter model for speculation to accelerate inference, and none of them consider the additional memory consumption for increased speed as an unjustifiable trade-off --  they don't even provide a comprehensive analysis of memory consumption as we did in Appendix B. Among them, **Deepmind's work [1] even proposed to employ a 4-billion parameter model as a drafter** to boost the inference speed of a 70B model. Moreover, even **GPT-4 is rumored to exploit our idea of speculative execution** to accelerate inference, according to the famous leak of the GPT-4 details (https://news.ycombinator.com/item?id=36710170).
>
> All the evidence clearly demonstrates that increasing memory consumption to enhance inference speed **is not an issue** in practice; otherwise, there would not be so many studies following this work to study the use of this approach to accelerate inference. Instead, we believe it will become **a highly practical solution and a key research direction** for optimizing inference speed in the future, as we suggested at the end of Appendix B.
>
> We sincerely hope our explanation **can resolve your concerns** regarding memory cost. We also hope that you can **reevaluate the value and significance** of our work, taking into consideration **the latest technological advancements and developments following our proposed idea of speculative execution**.
>
> &nbsp;
>
> ***Q2: It seems that the proposed approach introduces more computational overhead during inference. Is there any comparison about Carbon Emission? My concern is, even if the approach can generate faster, it actually consumes much more energy than before. I may save some money but actually, it is not that good for the environment.***
>
> R2: Thank you for your very interesting question about carbon emission. As you commented, SpecDec introduces extra computational overhead, which leads to an increase in GPU power consumption. However, **it can substantially reduce GPU hours owing to its high efficiency**. We compare GPU power consumption and GPU hours of autoregressive decoding and SpecDec for translating 3000 sentences in WMT14:
>
> | Decoding approaches                | GPU Type | GPU Avg. Power consumption (P)  | GPU-hours (t) | Total Energy consumption (E) | Carbon emitted (CO2eq) |
> | ---------------------- | -------- | --------------------- | --------- | ----------------------- | ---------------------- |
> | Autoregressive Decoding |  Nvidia P100     | 86W                   | 0.27h      | 25.54Wh                  | 9.83 g                  |
> | SpecDec                |  Nvidia P100     | 110W                   | 0.05h      | 6.05Wh                  | 2.33g                   |
>
> The total energy consumption and carbon emitted can be calculated as follows:
> $E=P \times t \times \text{PUE}$ where we set PUE (Power Usage Effectiveness) at 1.1. $\text{CO2eq}=0.385g*\text{CO2eq/Wh} \times E$ where 0.385g*CO2eq/Wh is the carbon intensity factor that is set based on the US national average.
>
> According to the above table, while SpecDec’s GPU power consumption is 28% higher, its GPU hours are **540% shorter than autoregressive decoding**. As a result, SpecDec’s total energy consumption is 23.7% of autoregressive decoding, resulting in **4.2x reduction of carbon emission**.
>
> Therefore, **SpecDec not only does not increase carbon emission but actually significantly reduces it, making it a more environmentally friendly option for inference.**
>
> We appreciate your suggestion and will add these details in the revised version to further strengthen our work's merits.

---

### Official Review · Reviewer_c6y8 · 2023-08-04

**Paper Topic And Main Contributions:** 1. The paper introduces a novel metho…
**Soundness:** 3

**Excitement:**

3: Ambivalent: It has merits (e.g., it reports state-of-the-art results, the idea is nice), but there are key weaknesses (e.g., it describes incremental work), and it can significantly benefit from another round of revision. However, I won't object to accepting it if my co-reviewers champion it.

**Missing References:**

Deep Compression: Compressing Deep Neural Networks with Pruning, Trained Quantization and Huffman Coding (ICLR 2016)
Eyeriss: An Energy-Efficient Reconfigurable Accelerator for Deep Convolutional Neural Networks (ISSCC 2016)
Extremely Fast Neural Machine Translation with Lightweight Convolutions" (EMNLP 2019)


**Questions For The Authors:**

1. Latency is one of the two major design principles claimed mentioned in this paper. However, there is no system profiling analysis to support this thrust. There are only a few latency numbers shown in the paper without discussions on what the computing platform is. What is the computing system configuration? GPU types? GPU numbers? GPU cache memory? CPU types & Numbers? Threads? Parallelism? Software package? Compiler support? Please provide the details.

2. This paper lacks a main discussion on the comparison with state-of-the-art acceleration techniques. Please provide quantitative analysis with other acceleration techniques, including weight pruning, head pruning, token pruning, knowledge distillation, and quantization. There are many works on each technique.

3. Due to the introduction of an additional Spec-Drafter module, SpecDec may incur additional memory costs at test time. Could you provide a detailed analysis and comparison of the memory overhead introduced by the Spec-Drafter module？eg. Deep Compression: Compressing Deep Neural Networks with Pruning, Trained Quantization and Huffman Coding (ICLR 2016)

4. The performance of SpecDec might be sensitive to the choice of hyperparameters, possibly adding to the complexity of tuning and optimization. Please investigate the impact of different hyperparameters on performance through a series of experiments.

5. Although SpecDec is suitable for a variety of tasks, its memory requirements may make it unsuitable for inference scenarios where memory is constrained. Please do some experiments to Study how SpecDec could be adapted to fit memory-constrained environments. Such as Eyeriss: An Energy-Efficient Reconfigurable Accelerator for Deep Convolutional Neural Networks (ISSCC 2016)

6. Could you do some In-depth analysis of the relationship between speed gains and potential quality loss. Extremely Fast Neural Machine Translation with Lightweight Convolutions" (EMNLP 2019)


**Reasons To Accept:**

1. SpecDec achieves significant speed improvements while maintaining a generation quality comparable to traditional AR methods. This balance makes it an appealing solution for various applications.
2. SpecDec demonstrates good performance across different seq2seq tasks and datasets, showing its wide applicability and robustness.
3. The paper clearly explains the workings of the method through charts and detailed descriptions, aiding in the understanding of the novel approach.


**Reasons To Reject:**

1. Due to the introduction of an additional Spec-Drafter module, SpecDec may incur additional memory costs at test time.
2. The performance of SpecDec might be sensitive to the choice of hyperparameters, possibly adding to the complexity of tuning and optimization.
3. Although SpecDec is suitable for a variety of tasks, its memory requirements may make it unsuitable for inference scenarios where memory is constrained.
3. This paper lacks a main discussion on the comparison with state-of-the-art acceleration techniques.
4. Latency is one of the two major design principles claimed mentioned in this paper. However, there is no system profiling analysis to support this thrust.


**Reproducibility:**

4: Could mostly reproduce the results, but there may be some variation because of sample variance or minor variations in their interpretation of the protocol or method.

**Reviewer Confidence:**

2: Willing to defend my evaluation, but it is fairly likely that I missed some details, didn't understand some central points, or can't be sure about the novelty of the work.

---

> ### Author Rebuttal · Authors · 2023-08-27
>
> Thank you for your valuable review comments. However, there is **a serious misunderstanding** regarding the memory cost issue and other concerns you raised. We hope our explanation and responses can resolve your misunderstanding and concerns, and improve your impression and evaluation of our paper:
>
> &nbsp;
>
> ***Q1: Due to the introduction of an additional Spec-Drafter module, SpecDec may incur additional memory costs at test time. Although SpecDec is suitable for a variety of tasks, its memory requirements may make it unsuitable for inference scenarios where memory is constrained.***
>
> R1: We have acknowledged in the limitations section that SpecDec introduces memory overhead as a trade-off for accelerating decoding. However, the benefits of our approach, including significant inference speedup and substantial improvements in the latency-throughput curve, **far outweigh the memory overhead**. As extensively discussed in Appendix B, the memory overhead introduced by SpecDec primarily stems from storing the weights of the draft model. Generally, this additional memory consumption is not excessive and does not increase significantly with the sequence length or batch size. Most importantly, memory is no longer the bottleneck for practical model deployment. With the emergence and maturity of **various data/tensor/pipeline parallelism techniques**, the addition of more GPUs **can easily address memory issues**, which is also why models continue to grow larger. In contrast, latency remains an inescapable bottleneck in model deployment that cannot be resolved merely by increasing the number of machines.
>
> **Precisely because memory is no longer the primary bottleneck in practical model deployment**, and the **urgency of addressing latency** issues has increased, there has been a growing interest in research **following our work to explore the methodology** of introducing a draft model to improve transformer inference. As **the first work** to propose the name "Speculative Decoding" and explicitly leverage the idea of speculative execution (You can consulate with SAC/AC to confirm this fact), **our approach has been acknowledged** by several recent papers studying this methodology:
>
> 1. Accelerating Large Language Model Decoding with **Speculative** Sampling. (Deepmind)
> 2. SpecTr: Fast **Speculative** Decoding via Optimal Transport. (Google Research)
> 3. SpecInfer: Accelerating Generative LLM Serving with **Speculative** Inference and Token Tree Verification
>
> **All these papers follow our work** to introduce a drafter model for speculation to accelerate inference, and none of them consider the additional memory consumption for increased speed as an unjustifiable trade-off --  they don't even provide a comprehensive analysis of memory consumption as we did in Appendix B. Among them, **Deepmind's work [1] even proposed to employ a 4-billion parameter model as a drafter** to boost the inference speed of a 70B model. Moreover, even **GPT-4 is rumored to exploit our idea of speculative execution** to accelerate inference, according to the famous leak of the GPT-4 details (https://news.ycombinator.com/item?id=36710170).
>
> All the evidence clearly demonstrates that increasing memory consumption to enhance inference speed **is not an issue** in practice; otherwise, there would not be so many studies following this work to study the use of this approach to accelerate inference. Instead, we believe it will become **a highly practical solution and a key research direction** for optimizing inference speed in the future, as we suggested at the end of Appendix B.
>
> We sincerely hope our explanation **can resolve your concerns** regarding memory cost. We also hope that you can **reevaluate the value and significance** of our work, taking into consideration **the latest technological advancements and developments following our proposed idea of speculative execution**.
>
> &nbsp;
>
> ***Q2: Latency is one of the two major design principles mentioned in this paper. However, there is no system profiling analysis to support this thrust. There are only a few latency numbers shown in the paper without discussions on what the computing platform is. What is the computing system configuration? GPU types? GPU numbers? GPU cache memory? CPU types & Numbers? Threads? Parallelism? Software package? Compiler support? Please provide the details.***
>
> R2: Thank you for your comments. As we introduced in Section 4.1, we use 1 Nvidia P100 GPU based on fairseq implementation in our evaluation. We mainly report speedup instead of absolute latency because absolute latency is very sensitive to the computing devices and runtime. In contrast, the speedup is more insensitive. As we report in Table 1, Blockwise Decoding’s speedup is almost the same in both its original paper and our reimplementation. Therefore, we skip other details of computing devices and runtime. Specifically, the machine we mainly use for evaluation is **Azure Standard_NC24s_v2** (https://learn.microsoft.com/en-us/azure/virtual-machines/ncv2-series). It has 24 vCPU (Intel Xeon E5-2690 v4), 448GB memory. We use Pytorch 1.10 with cuda 11 as our runtime. **We are committed to adding the details in the revised version**.
>
> &nbsp;
>
> ***Q3: This paper lacks a main discussion on the comparison with state-of-the-art acceleration techniques. Please provide quantitative analysis with other acceleration techniques, including weight pruning, head pruning, token pruning, knowledge distillation, and quantization. There are many works on each technique.***
>
> R3: While we acknowledge the relevance of various acceleration methods, such as weight pruning, head pruning, token pruning, knowledge distillation, and quantization, we position SpecDec as a novel decoding algorithm that is **orthogonal to these techniques**. Therefore, our main baselines and comparable counterparts are **those decoding approaches such as Autoregressive Decoding and Blockwise Decoding**.
>
> We understand your interest in seeing the comparison to techniques of other categories (e.g., pruning, quantization, or distillation). However, we hope you can understand that it is a common practice for **a paper focusing on a specific acceleration category not to compare with techniques from other categories**. This is because different categories of acceleration techniques often address different aspects of model optimization, and directly comparing them might not provide meaningful insights into the specific improvement brought by the proposed method in its targeted category. For example, you would **rarely see** a paper on quantization comparing its results to pruning methods, a paper on pruning comparing its results to distillation methods, or a paper on distillation comparing its results to decoding algorithms. Some recent examples include:
>
> 1. Quantization: "Quantization and Training of Neural Networks for Efficient Integer-Arithmetic-Only Inference" (CVPR 2018) - This paper focuses on comparing different **quantization** methods and **does not compare with pruning or distillation techniques**.
> 2. Pruning: "The Lottery Ticket Hypothesis: Finding Sparse, Trainable Neural Networks" (ICLR 2019) - This paper compares various **pruning** approaches and **does not compare with distillation or quantization techniques**.
> 3. Distillation: "DistilBERT, a distilled version of BERT: smaller, faster, cheaper and lighter" (NeurIPS 2019) - This paper focuses on **knowledge distillation** methods and **does not compare with pruning or quantization techniques**.
> 4. Decoding algorithm: "Blockwise Parallel Decoding for Deep Autoregressive Models" (NeurIPS 2021) - This paper presents a **decoding algorithm** and compares it with other decoding approaches, but **does not compare with pruning, distillation, or quantization methods**.
>
> We hope this clarification helps in understanding **our choice of baselines and comparable counterparts** for our proposed SpecDec method.
>
> &nbsp;
>
> ***Q4: The performance of SpecDec might be sensitive to the choice of hyperparameters, possibly adding to the complexity of tuning and optimization. Please investigate the impact of different hyperparameters on performance through a series of experiments.***
>
> R4: Our submission has **already provided detailed ablation studies and analyses of various hyperparameters in Sections 4.3.1 and 4.3.2** of the paper. Please refer to **Table 3 and Table 4** in Section 4.3.1 and 4.3.2 for the details.
>
> **It is not true** that it adds to the complexity of tuning and optimization. SpecDec only introduces 3 hyperparameters. Only k is involved during training, and it can be **safely set to 20~30 (see Table 3)**.
> For beta and tau, **they are only used during inference**, like the beam b in beam search. Rather than increasing the tuning difficulty, SpecDec provides a solution to any quality-speed trade-off. By adjusting these hyperparameters, users can easily find a balance between the desired acceleration and the quality of the generated outputs. **This flexibility allows SpecDec to be adaptable to various application scenarios, where different speed and quality requirements might be needed.**
>
> &nbsp;
>
> ***Q5: Could you do some In-depth analysis of the relationship between speed gains and potential quality loss?***
>
> R5: We have **already performed a detailed analysis of the speedup-quality tradeoff of SpecDec in Table 4 of the manuscript**. As Table 4 clearly demonstrates, as AR verification becomes less strict (i.e., the beta and tau increase), more tokens (Tok.) will be accepted and the speedup of SpecDec will accordingly increase. When AR verification becomes too relaxed, there will be a risk of degradation in quality. Therefore, as long as we do not make AR verification too relaxed (high beta and tau), **SpecDec can achieve significant speedup with almost no loss in generation quality**. This is the main difference between SpecDec and previous methods that lead to a loss in generation quality. Please refer to Table 4 and 5 for more details.

---

### Official Review · Reviewer_u3hL · 2023-08-05

**Soundness:** 4

**Excitement:**

3: Ambivalent: It has merits (e.g., it reports state-of-the-art results, the idea is nice), but there are key weaknesses (e.g., it describes incremental work), and it can significantly benefit from another round of revision. However, I won't object to accepting it if my co-reviewers champion it.

**Paper Topic And Main Contributions:**

This paper describes how to reduce the latency of AR decoding (especially in seq2seq format.)
They suggest draft-and-verify speculative decoding.
What they contributed are
1. Accurate draft by Masked Language Modeling in decoder conditioned by a few leftward contexts
2. Fast draft by reducing the number of decoder layers and reallocating them to the encoder side.
3. Alleviating the rigid verify that only checks top-1 AR result to top-beta so verify wouldn't be an obstacle to fast decoding.

**Questions For The Authors:**

1. Is this method also applicable to decoder-only models such as GPT or Llama?

**Reasons To Accept:**

1. The paper is very well written and well structured. It is easy to understand what they did if you have enough knowledge on Transformers.
2. Changing the AR decoding side to be NAR (or semi-AR) and increasing a fast encoder's burden makes sense.
3. Combining the 3 contributions I wrote above is a novel approach.
4. The results looks promising as it shows around 5 times speed up

**Reasons To Reject:**

1. There's a lack of comparison to other fast decoding algorithms such as "early-exiting" (Confident Adaptive Language Modeling, Shuster, 2022), or "speculative decoding" which uses one small transformer (draft) and one larger transformer (verify). Please refer to "Fast Inference from Transformers via Speculative Decoding (Leviathan et al 2023), and "Accelerating Large Language Model Decoding with Speculative Sampling" (Chen et al 2023) for speculative decoding.
2.  It is hard to implement the SpecDec of this paper because the method needs training while other methods usually only need to infer the existing models.
3. It is not sure that this would work on a larger scale such as a model larger than or around 10B (scalability). They only tried very small models (12 enc, 2 dec) Would this be also 5X speed up at 10B or 100B models?

**Reproducibility:**

3: Could reproduce the results with some difficulty. The settings of parameters are underspecified or subjectively determined; the training/evaluation data are not widely available.

**Reviewer Confidence:**

4: Quite sure. I tried to check the important points carefully. It's unlikely, though conceivable, that I missed something that should affect my ratings.

---

> ### Author Rebuttal · Authors · 2023-08-27
>
> Thank you for your constructive review comments. We appreciate your recognition of our work (very well written + novel and sound method + promising and significant results). We understand your questions and concerns in your review comments, but we believe **these minor issues should not overshadow our merits and contributions (with a score of 3)**. We hope that our following responses will help you better interpret our **merits, originality, and significance** and resolve your **misunderstanding**.
>
> &nbsp;
>
> ***Q1: There's a lack of comparison to other fast decoding algorithms such as "early-exiting" (Confident Adaptive Language Modeling, Shuster, 2022), or "speculative decoding" which uses one small transformer (draft) and one larger transformer (verify). Please refer to "Fast Inference from Transformers via Speculative Decoding (Leviathan et al 2023), and "Accelerating Large Language Model Decoding with Speculative Sampling" (Chen et al 2023) for speculative decoding.***
>
> R1: As you mentioned Leviathan et al (2023) and Chen et al (2023), we believe you must be aware of the methodology’s significance and **the surge in the Speculative Decoding/Sampling/Inference research** in the recent year:
>
> [1]. Leviathan et al (2023): Fast Inference from Transformers via Speculative Decoding
>
> [2]. Chen et al (2023): Accelerating Large Language Model Decoding with Speculative Sampling
>
> [3]. SpecTr: Fast Speculative Decoding via Optimal Transport
>
> [4]. SpecInfer: Accelerating Generative LLM Serving with Speculative Inference and Token Tree Verification
>
> [5] . ...
>
> However, we believe you may **misunderstand** that we followed Leviathan et al (2023) and Chen et al (2023). Instead, the fact is that **this work is much earlier than Leviathan et al (2023) and Chen et al (2023). It is the first work** that proposes the name "Speculative Decoding" which explicitly **leverages the idea of speculative execution** to accelerate decoding. It is **the origin of** the above Speculative Decoding/Sampling/Inference research surge. Both Leviathan et al (2023) and Chen et al (2023) as well as other above papers **followed and cited** this work (You can consult with the AC/SAC to confirm this fact). We hope this can address your misunderstanding in terms of this work's originality and significance and **reconsider your score on this work based on the context that it is prior to Leviathan et al (2023) and Chen et al (2023)**.
>
> Although **this work indeed predates** Shuster et al (2022), Leviathan et al. (2023) and Chen et al. (2023), we understand the importance of benchmarking our approach against these more recent methods for a comprehensive evaluation, as you suggested. To address your comment and further demonstrate the effectiveness of our method, **we have conducted additional experiments comparing our SpecDec approach with the mentioned algorithms**:
>
> The following results are the comparison of our reimplemented Leviathan et al. (2023) and ours in accelerating T5-XXL (11B) for WMT EN-DE translation:
>
> | Decoding algorithm (T5-XXL) | BLEU      | Speed    |
> | :--------------------------------------- | :-------- | :------- |
> | Autoregressive Decoding         | 32.23     | 1.0x     |
> | Leviathan et al (0.1B drafter, T5-Small) | 32.23     | 2.8x     |
> | Leviathan et al (0.8B drafter, T5-Large) | 32.23     | 1.6x     |
> | Ours (0.5B drafter)                   | **32.31** | **4.9x** |
>
> | Models                 | Enc,Dec | d_model | d_ff | #heads | #Params |
> | :--------------------- | :------ | :------ | :--- | :----- | :------ |
> | SpecDec (0.5B drafter) | 24,6    | 1024    | 4096 | 16     | 0.5B    |
>
> Our approach uses a 0.5B Spec-Drafter to achieve an almost 5x speedup without quality degradation, while Leviathan et al. (2023)'s speedup results are lower than 3x. The reasons for our approach significantly outperforming Leviathan et al. (2023) are **twofold**:
>
> 1. Our Spec-Drafter is specially learned to draft for accelerating the target model with the speculation execution idea: its draft results are **better aligned** with the target model's results.
> 2. Our approach uses a fast deep-encoder-shallow-decoder architecture and a non-autoregressive approach to generate drafts, which is significantly **more efficient** than their autoregressive drafting method.
>
> The following results are the comparison of our reimplemented Shuster et al. (2022) and ours in accelerating a Transformer-base model for WMT DE-EN translation:
>
> | Models               | BLEU      | Speed    |
> | :------------------- | :-------- | :------- |
> | Transformer-base     | 28.89     | 1.0x     |
> | Shuster et al. (2022) | 28.16     | 2.4x     |
> | SpecDec              | **28.93** | **5.1x** |
>
> While Shuster et al. (2022) achieve 2.4x speedup, it suffers from serious quality degradation. In contrast, our SpecDec can achieve 5.1x speedup without quality degradation, **showing the superiority over Shuster et al. (2022) in terms of both generation quality and speedup**.
>
>
> &nbsp;
>
> ***Q2: It is hard to implement the SpecDec of this paper because the method needs training while other methods usually only need to infer the existing models.***
>
> R2: All SpecDec methods, including this work, Leviathan et al (2023), Chen et al (2023) and SpecInfer[5], require a draft model for speculation to speed up the process. However, **it is not true** that other methods only need to infer the existing models -- The fact is **only Leviathan et al. (2023) employ the existing models as a drafter**. As we show the results in our response to Q1, **using an existing model as a drafter is not the best choice** because existing models **fall short in terms of both draft quality and draft efficiency** compared to a specially designed drafter for speculation. This is also **the reason why not only our work but also Chen et al (2023) and SpecInfer[5] use a specialized model for drafting**: e.g., Chen et al (2023) specially train **a 4B model with only 8 layers** -- a shallow and wide model -- as a drafter, as shallow and wide models have higher draft efficiency. (FYI, existing models at the 4B scale are typically much deeper -- e.g., 32 layers or more.)
>
> Moreover, even the GPT-4 is rumored to exploit the idea of speculative execution to accelerate inference, according to the famous leak of the GPT-4 details (https://news.ycombinator.com/item?id=36710170). If this rumor is true, we believe its drafter model would not be an existing model (deep and slow); instead, it is more likely to be a specialized drafter model that is fast and better aligned with the target model.
>
> The feasibility, reliability, and practicality of training a new model for drafting have already been corroborated by the multiple papers mentioned above. We believe that **offline training of a specialized drafter model to better accelerate the online inference** has significant practical significance and value. Our comprehensive **opensourced** code, data, and models can help anyone easily implement our idea for easy reproduction.
>
> &nbsp;
>
> ***Q3: It is not sure that this would work on a larger scale such as a model larger than or around 10B (scalability). They only tried very small models (12 enc, 2 dec) Would this be also 5X speed up at 10B or 100B models?***
>
> R3: This work tests on 6+6-512 (Transformer-base), 6+6-1024 (Transformer-big), 12+2-512/1024, and BART for fair comparison because most previous work on seq2seq generation acceleration reports results with these model sizes. **Our approach can of course scale to large models**:
>
> Our results in our response to Q1 already indicate that **when applied to 11B models, SpecDec can also achieve a promising 5x speedup** even with a much smaller Spec-Drafter (0.5B), demonstrating that **the effectiveness of SpecDec is not affected by the model size.**
>
> These results are also consistent with the observations of Chen et al. (2023), which uses the speculation idea to accelerate a 70B decoder-only model, confirming that the speculative execution methodology's effectiveness is not affected by the model size. **We'll add these results in the revised version.**
>
> &nbsp;
>
> ***Q4: Is this method also applicable to decoder-only models such as GPT or Llama?***
>
> R4: The target model to be accelerated by SpecDec **can be both enc-dec and dec-only** architectures because **the target model is used as the verifier in SpecDec and it is forwarded in the same way as it is trained**. However, it is notable that the Spec-drafter in our submission that uses a non-autoregressive model better fits seq2seq tasks for maximizing the acceleration effect. When applied to non-seq2seq tasks like next-word prediction, the Spec-drafter may need to be accordingly adjusted.

---

### Meta-Review · Area_Chair_7E2m · 2023-09-18

**Recommendation:** 3

**Metareview:**

This paper proposes a speculative decoding approach for LLMs by using a separate smaller encoder-decoder drafting model to quickly generate multiple tokens and then verify these tokens in parallel using a larger model. The paper is generally well written, has a clearly motivated idea, and is a part of an emerging research area.

The framework of speculative decoding has seen a resurgence in interest over the 18 months (as indicated by the authors in the comments below). One of the reviewers mentions several other speculative decoding works that have appeared in that time frame were not compared against. The authors provide additional extensive results in the rebuttal compared against these other works. The results suggest that their approach outperforms other existing approaches.

Two reviewers raised concerns around the memory/computational overhead of requiring two models (a drafting model and validation model) compared to the conventional approach. In the rebuttal, the reviewers provide additional results to show that Speculative Decoding requires more power but requires less energy due to the increased tokens per second of the draft-then-verify approach. Generally, I think this issue of how this approach scales to larger models is something that could be further explored. For instance, how would this comparison look under a setting where both the drafter and verification models could not both fit in GPU memory? My assumption is that IO for weight transfer would then become a limiting factor.

---

### Decision · Program_Chairs · 2023-10-07

**Decision:**

Accept-Findings

**Comment:**

This paper proposes a speculative decoding approach for LLMs by using a separate smaller encoder-decoder drafting model to quickly generate multiple tokens and then verify these tokens in parallel using a larger model. The paper is generally well written, has a clearly motivated idea, and is a part of an emerging research area.

The framework of speculative decoding has seen a resurgence in interest over the 18 months (as indicated by the authors in the comments below). One of the reviewers mentions several other speculative decoding works that have appeared in that time frame were not compared against. The authors provide additional extensive results in the rebuttal compared against these other works. The results suggest that their approach outperforms other existing approaches.

Two reviewers raised concerns around the memory/computational overhead of requiring two models (a drafting model and validation model) compared to the conventional approach. In the rebuttal, the reviewers provide additional results to show that Speculative Decoding requires more power but requires less energy due to the increased tokens per second of the draft-then-verify approach. Generally, I think this issue of how this approach scales to larger models is something that could be further explored. For instance, how would this comparison look under a setting where both the drafter and verification models could not both fit in GPU memory? My assumption is that IO for weight transfer would then become a limiting factor.